# Chronic systemic inflammation predicts long-term mortality among patients with fatty liver disease: Data from the National Health and Nutrition Examination Survey 2007–2018

Xinghe Pan[1☯], Jie Lv[2☯], Man Liu[3☯], You Li[1☯], Yitong Zhang[1], Rui Zhang[1], Junliang Liu[1], Chenglin Sun[1‡]*, Hongpeng Guo[1‡]*

1 Department of General Surgery, Central Hospital Affiliated to Shenyang Medical College, Shenyang, China, 2 Department of Clinical Laboratory Center, Xinjiang Medical University Affiliated Tumor Hospital, Urumqi, China, 3 School of Basic Medical Sciences, Shenyang Medical College, Shenyang, China

☯ These authors contributed equally to this work.
‡ These authors also contributed equally to this work
* scl9999@163.com (CS); 17642014915@163.com (HG)

long-term mortality among patients with fatty liver disease: Data from the National Health and Nutrition Examination Survey 2007–2018. PLoS ONE 19(11): e0312877. https://doi.org/10.1371/journal.pone.0312877

## Abstract

### Background

Low-grade systemic inflammation (SI) in patients with fatty liver disease (FLD) is an important hallmark of disease onset and progression. This study aims to evaluate the prognostic significance of novel SI markers in FLD.

### Methods

This was a retrospective cohort study. We included adult patients with FLD with complete data and analyzed the association between chronic SI and long-term mortality in patients with FLD. Systemic immune-inflammation index (SII), pan-immune-inflammation value (PIV), and systemic inflammation response index (SIRI) were evaluated based on peripheral blood counts and FLD was determined by the Fatty Liver Index.

### Results

A total of 5497 patients with FLD were included in the final analysis. SII and PIV (but not SIRI) were found to be associated with all-cause and cardiovascular mortality in univariate analysis. Multivariate Cox regression analysis and KM analysis demonstrated that SII and PIV were associated with all-cause mortality, with SII showing a nonlinear correlation in RCS. PIV (but not SII) was associated with the cardiovascular-related survival probability over time. Stratified analysis indicated that the positive correlation between SII and PIV and all-cause mortality was not altered by subgroups.

### Conclusions

SII and PIV are strongly and consistently associated with all-cause mortality in patients with FLD, with PIV potentially showing a closer association with cardiovascular mortality.

**Data Availability Statement:** All relevant data are within the manuscript and its Supporting information files.

**Funding:** This study were supported under the Liaoning Provincial Science and Technology Plan project (Grant no.2022JH2/1013),the Shenyang Science and Technology Plan Project(Grant no.21173918),the Graduate Student Science and Technology Innovation Fund of Shenyang Medical College (Grant no. Y20220531) and Shenyang Health Commission Scientific Research Project (Grant no. 202358). The funders had no role in study design, data collection and analysis, decision to publish, or preparation of the manuscript.

**Competing interests:** The authors have declared that no competing interests exist.

## Introduction

Fatty liver disease (FLD) is a liver metabolic disease characterized by hepatic steatosis, which can be classified into nonalcoholic fatty liver disease (NAFLD), alcoholic fatty liver disease (AFLD), and the recently proposed metabolic-associated fatty liver disease (MAFLD) according to the etiology [1,2]. Nonalcoholic fatty liver disease (NAFLD) is currently the most prevalent chronic liver disease worldwide, with an estimated global prevalence of 30% [3,4]. AFLD and MAFLD also present significant global public health concerns [5,6]. Therefore, FLD is a major contributor to liver-related morbidity and mortality, and improved prediction and prognosis of FLD are essential to alleviate this significant disease burden.

Chronic systemic inflammation (SI) is a key hallmark of various metabolic disorders including obesity, diabetes, and FLD [7]. Hepatic lipotoxicity and adipose tissue dysfunction in FLD trigger downstream pathophysiological processes, including mitochondrial dysfunction, endoplasmic reticulum stress, and oxidative stress. These pathological conditions can induce local inflammation and SI and contribute to the activation of the innate immune system, including recruitment of pro-inflammatory cells and pathways and the release of pro-inflammatory cytokines [8]. Accumulating evidence indicates that SI and dysregulated immune response drive substantial disease progression in FLD and therefore may have prognostic value [9–11].

Various peripheral blood cell-based SI biomarkers have been shown to possess predictive and prognostic value in FLD. Recently, several novel SI indices such as SII (systemic immune-inflammation index), PIV (pan-immune-inflammation value), and SIRI (systemic inflammation response index) have been proposed to provide prognostic significance in various diseases. SII has been shown to be significantly associated with all-cause mortality in patients with atherosclerotic cardiovascular disease [12]. Another study using the National Health and Nutrition Examination Survey (NHANES) database, which included 42,875 US adults with a 20-year follow-up indicated that SII and SIRI were strongly associated with all-cause and cause-specific mortality in the general population [13]. Furthermore, SIRI and PIV have also been associated with all-cause mortality and cardiovascular mortality among hypertensive adults [14,15]. SII was shown to be correlated with liver steatosis and fibrosis in NAFLD according to a recent NHANES study, suggesting that SI markers may be correlated with disease progression in NAFLD [16,17]. However, the prognostic value of these novel SI biomarkers in patients with FLD, particularly their association with mortality, remains unexplored.

These simple and readily available peripheral blood biomarkers may hold prognostic value for the FLD population in clinical practice. Therefore, we aimed to investigate the association of these SI metrics with all-cause and cardiovascular mortality in a nationally representative population-based study here.

## Methods

### Study design

NHANES is a nationally representative, multistage, cross-sectional survey conducted by the National Center for Health Statistics (NCHS) and is designed to assess the health and nutrition status of US non-institutionalized populations. The NCHS Ethics Review Board approved all NHANES study protocols, and all participants provided written informed consent. We utilized data from NHANES 2007–2018 (including 6 continuous cycles, 2007–2008, 2009–2010, 2011–2012, 2013–2014, 2015–2016, 2017–2018) for this study. Baseline FLD participants from NHANES 2007–2018 were followed until December 31, 2019, and mortality data was obtained by prospectively matching with death certificate information from the Public-use Linked

Mortality Files of the National Death Index (NDI) database (https://www.cdc.gov/nchs/data-linkage/mortality-public.htm) [13,18]. The study inclusion criteria included FLD patients aged 18 years or older with complete data. Exclusion criteria were participants with missing data and extreme SI values and a follow-up duration of less than 24 months.

### Definition of SI markers

These SI markers were calculated using neutrophils, lymphocytes, monocytes, and/or platelets derived from complete blood counts (CBC) as measured by automated hematology analyzers [19]. The following formulas were used to calculate SI indicators: SII = (platelet count × neutrophil count)/lymphocyte count [20]; SIRI = (neutrophil count × monocyte count)/lymphocyte count [21]; PIV = (platelet count × neutrophil count × monocyte count)/lymphocyte count [22].

### Definition of FLD

We used the Fatty Liver Index (FLI) to define hepatic steatosis, which has been validated as a simple and accurate indicator for determining liver fat accumulation in real-world settings (FLI ≥ 60 indicates hepatic steatosis) [23]. FLI was calculated using serum triglycerides (TG), body mass index (BMI), waist circumference (WC), and serum gamma-glutamyl transpeptidase (GGT). FLI = $(e^{0.953*\log e\ (TG) + 0.139*BMI + 0.718*\log e\ (GGT) + 0.053*WC-15.745})$ / $(1 + e^{0.953*\log e\ (TG) + 0.139*BMI + 0.718*\log e\ (GGT) + 0.053*WC-15.745}) \times 100$.

### Outcome ascertainment

The primary outcomes of the study were all-cause mortality and cardiovascular mortality. Mortality data were obtained through linkage with the NDI. The mortality from cardiovascular disease (CVD) was classified according to ICD-10 codes I00-I09, I11, I13, and I20-I51. All included patients with FLD were followed until December 31, 2019.

### Determination of covariates

Several key confounding factors were selected, including age, sex, ethnicity, education level, family income-to-poverty ratio (PIR), and marital status, smoking, alcohol consumption, physical work, BMI, WC, total cholesterol, high-density lipoprotein (HDL)-cholesterol, alanine transaminase (ALT), aspartate transaminase (AST), hypertension, and diabetes. Smoking status was categorized as follows: (1) Never: individuals who answered that they had smoked fewer than 100 cigarettes in life; (2) Former: have smoked at least 100 cigarettes throughout their lives, but are currently not smoking; (3) Current: had smoked at least 100 cigarettes in life and smoked at present [24]. Alcohol consumption was classified as follows: (1) Never: alcohol consumption < 12 times per lifetime; (2) Former: at least 12 drinking occasions within a year or lifetime and no drinking in the last year (3) Heavy: at least 3 drinks/d for women/4 drinks/d for men, or binge drinking ≥ 5 d/month; (4) Moderate: at least 2 drinks/d for women/3 drinks/d for men, or binge drinking ≥ 2 d/month; (5) Mild: fewer than 2 drinks/d [25]. For physical work, the diagnostic criteria were based on the descriptions in the NHANES: (1) No: no work activities; (2) Mild strength: causing small increases in respiration or heart rate; (3) Medium to high strength: causing large increases in respiration or heart rate or the above situation exists. Hypertension was diagnosed based on self-reported history, taking anti-hypertensive medication, or blood pressure values were within the hypertension range [26]. Diabetes was diagnosed if a physician indicates that a person has diabetes, blood glucose test suggesting diabetes, or taking diabetes medication [27].

## Statistical analysis

Our study was conducted using EmpowerStats and R software. All analyses were conducted appropriately according to NHANES guidelines [28]. In the baseline analysis, continuous variables were analyzed using one-way analysis of variance (ANOVA), and categorical variables were analyzed using the chi-square test. We initially conducted univariate analysis and subsequently constructed multivariate Cox proportional hazards regression models to assess the effects of SII and PIV on all-cause and cardiovascular mortality. For all-cause mortality, the crude model was unadjusted, and the adjusted model was adjusted for all important covariates in the univariate analysis including age, BMI, PIR, sex, ethnicity, marital status, education level, WC, ALT, total cholesterol, smoking, alcohol consumption, physical work, hypertension, and diabetes. For cardiovascular mortality, the crude model was unadjusted, and the adjusted model was adjusted for all significant covariates including age, PIR, total cholesterol, HDL-cholesterol, physical work, hypertension, and diabetes.

Kaplan-Meier (KM) survival curves and log-rank tests were used to assess differences in all-cause and cardiovascular survival among FLD participants across low, medium, and high SI marker levels. Restricted cubic splines (RCS) were used to examine the potential nonlinear correlations. The curve fitting term was defined using the RCS function from the rms package, and the degrees of freedom (or knots) are determined according to the magnitude of the *P* for nonlinear value [29]. Stratified analyses were performed to explore whether these associations remained stable across subgroups and to identify factors that may significantly influence these associations through interaction analyses. In sensitivity analyses, we first explored the association of other inflammatory markers such as C-reactive protein (CRP) with mortality in patients with FLD and the correlation of CRP with SII/PIV. Second, considering the potential bias introduced by excluding data with follow-up length less than 24 months and extreme data on SI markers, we did not exclude this group of patients in the sensitivity analysis to verify the stability of the conclusions. Finally, we explored the threshold effect of the nonlinear association between SII with all-cause mortality in patients with FLD. A P-value of less than 0.05 was considered statistically significant in all analyses.

## Results

### Participants

We initially enrolled 57,414 individuals aged 18 years and older and excluded those without diagnostic indicators for FLD (N = 40,125). Next, we excluded individuals with missing mortality data (N = 2325), missing blood biochemical tests data (N = 54), missing covariate data such as marital status (N = 721), education level (N = 14), hypertension (N = 137), smoking (N = 12), and physical work (N = 2). Finally, from the population with complete data, we excluded extreme values of SI markers (N = 70), those without FLD according to the diagnostic criteria (N = 7832), and patients who died within 24 months of follow-up (N = 626). Therefore, 5497 patients with FLD (2954 men and 2543 women) were included in the current study (**Fig 1**). The median follow-up duration for FLD patients was 88 months(interquartile range: 55–121 months).

### Baseline characteristics

**Table 1** described the baseline characteristics of patients grouped according to SII tertiles (≤ 370.9, 370.9 ~ 571.8, and ≥ 571.8). BMI, WC, ALT, AST, SII, PIV, SIRI, sex, and ethnicity showed significant differences between groups (all *P*-values for trend < 0.0001). Statistically significant trends were also observed for total cholesterol, hypertension, diabetes, and smoking

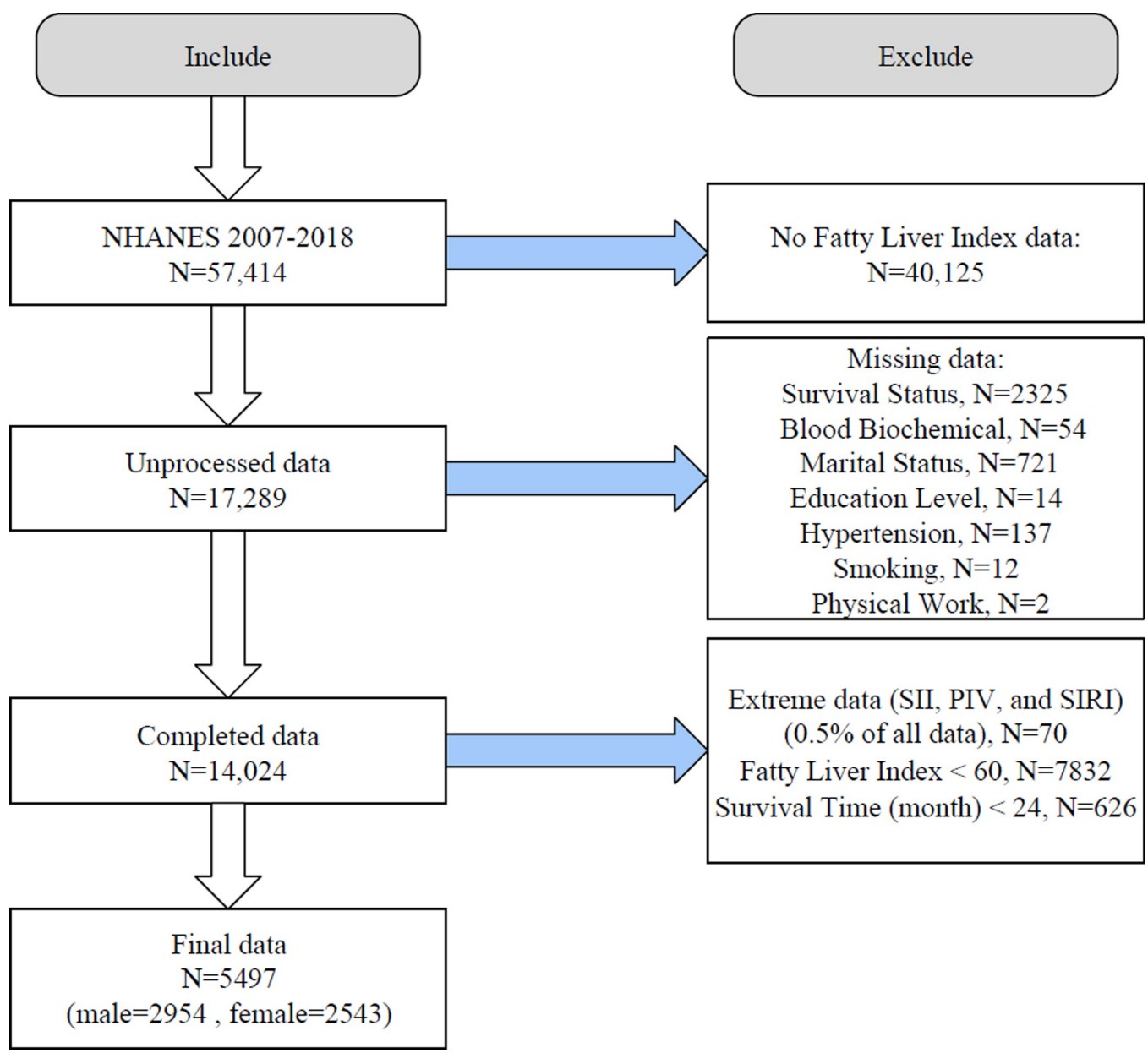

**Fig 1. Study subject inclusion and exclusion flowchart, NHANES 2007–2018.**

(all *P*-values for trend < 0.05). However, no significant differences were observed between groups in age, PIR, HDL-cholesterol, marital status, education level, alcohol consumption, and physical work (all *P*-values for trend > 0.05). **Table 2** presented the baseline characteristics of patients grouped according to PIV tertiles (≤ 184.4, 184.4 ~ 313.8, and ≥ 313.8). Significant differences were observed in BMI, SII, PIV, SIRI, ethnicity, and smoking (all *P*-values for trend < 0.0001), and in age, ALT, total cholesterol, HDL-cholesterol, education, hypertension, and diabetes between groups (all *P*-values for trend < 0.05). However, no significant differences were found for PIR, AST, sex, marital status, alcohol consumption, and physical work (all *P*-values for trend > 0.05). Baseline analysis grouped by tertile of SIRI was presented in **S1 Table**.

**Table 1. Baseline characteristics of patients according to SII tertile.**

| Variable | ≤ 370.9 | 370.9 ~ 571.8 | ≥ 571.8 | P-values for trend |
|---|---|---|---|---|
| Age, years | 49.46 (0.53) | 49.70 (0.44) | 50.12 (0.53) | 0.64 |
| Sex | | | | **< 0.0001** |
| Female | 727 (36.28) | 836 (43.00) | 980 (52.13) | |
| Male | 1105 (63.72) | 996 (57.00) | 853 (47.87) | |
| Ethnicity | | | | **< 0.0001** |
| Mexican American | 371 (12.39) | 395 (11.66) | 314 (9.51) | |
| Non-Hispanic Black | 453 (13.82) | 335 (9.40) | 264 (7.29) | |
| Non-Hispanic White | 642 (61.69) | 776 (68.64) | 922 (71.78) | |
| Other Hispanic | 227 (6.69) | 197 (5.27) | 224 (6.54) | |
| Other races | 139 (5.41) | 129 (5.03) | 109 (4.89) | |
| Marital status | | | | 0.54 |
| Single | 689 (34.46) | 668 (32.81) | 708 (32.48) | |
| Non-single | 1143 (65.54) | 1164 (67.19) | 1125 (67.52) | |
| Education | | | | 0.1 |
| < high school | 799 (39.65) | 784 (41.85) | 741 (37.63) | |
| high school | 704 (36.11) | 712 (33.19) | 757 (39.04) | |
| > high school | 329 (24.24) | 336 (24.95) | 335 (23.33) | |
| BMI, kg/m$^2$ | 33.35 (0.17) | 34.22 (0.21) | 35.30 (0.21) | **< 0.0001** |
| PIR | 2.87 (0.05) | 2.97 (0.06) | 2.85 (0.07) | 0.19 |
| WC, cm | 111.02 (0.39) | 112.82 (0.45) | 114.96 (0.42) | **< 0.0001** |
| ALT, U/L | 33.83 (0.80) | 30.92 (0.60) | 27.52 (0.58) | **< 0.0001** |
| AST, U/L | 29.33 (0.59) | 26.36 (0.36) | 26.04 (0.63) | **< 0.0001** |
| Total cholesterol, mmol/L | 5.17 (0.04) | 5.17 (0.04) | 5.05 (0.03) | **0.01** |
| HDL-cholesterol, mmol/L | 1.21 (0.01) | 1.20 (0.01) | 1.22 (0.01) | 0.46 |
| SII | 281.89 (1.79) | 461.53 (1.56) | 834.84 (9.02) | **< 0.0001** |
| PIV | 152.22 (2.10) | 254.73 (3.23) | 502.41 (8.61) | **< 0.0001** |
| SIRI | 1291.52 (14.01) | 1860.58 (25.02) | 2748.48 (38.38) | **< 0.0001** |
| Drinking | | | | 0.43 |
| Never | 228 (10.61) | 220 (10.34) | 230 (11.66) | |
| Former | 315 (17.10) | 325 (15.61) | 321 (17.98) | |
| Mild | 539 (34.88) | 509 (35.44) | 526 (33.77) | |
| Moderate | 193 (13.58) | 243 (16.90) | 235 (15.11) | |
| Heavy | 385 (23.83) | 379 (21.72) | 372 (21.48) | |
| Smoking | | | | **0.03** |
| No | 994 (53.75) | 982 (54.43) | 885 (48.04) | |
| Former | 507 (28.85) | 507 (27.84) | 535 (30.94) | |
| Current | 331 (17.41) | 343 (17.73) | 413 (21.02) | |
| Physical work | | | | 0.28 |
| No | 1074 (52.02) | 1029 (52.19) | 1074 (54.64) | |
| Mild | 364 (23.35) | 428 (26.02) | 403 (24.35) | |
| Medium to high | 394 (24.63) | 375 (21.79) | 356 (21.01) | |
| Hypertension | | | | **0.01** |
| No | 850 (49.39) | 882 (50.85) | 769 (44.22) | |
| Yes | 982 (50.61) | 950 (49.15) | 1064 (55.78) | |
| Diabetes | | | | **0.02** |
| No | 1269 (75.73) | 1255 (73.31) | 1186 (70.21) | |
| Yes | 563 (24.27) | 577 (26.69) | 647 (29.79) | |

Abbreviations: BMI, body mass index; WC, waist circumference; PIR, family income-to-poverty ratio; HDL, high-density lipoprotein; ALT, alanine transaminase; AST, aspartate transaminase. Continuous variables were described as mean (standard deviation) and number (percentage) was used to indicate categorical variables.

**Table 2. Baseline characteristics of patients according to PIV tertile.**

| Variable | ≤ 184.4 | 184.4 ~ 313.8 | ≥ 313.8 | P-values for trend |
|---|---|---|---|---|
| Age, years | 49.39 (0.51) | 49.05 (0.47) | 50.83 (0.49) | **0.02** |
| Sex | | | | 0.42 |
| Female | 845 (43.22) | 825 (43.33) | 873 (45.86) | |
| Male | 987 (56.78) | 1007 (56.67) | 960 (54.14) | |
| Ethnicity | | | | **< 0.0001** |
| Mexican American | 400 (13.70) | 377 (10.92) | 303 (9.11) | |
| Non-Hispanic Black | 496 (15.49) | 304 (8.21) | 252 (7.04) | |
| Non-Hispanic White | 571 (59.09) | 822 (69.87) | 947 (72.76) | |
| Other Hispanic | 224 (6.43) | 205 (5.81) | 219 (6.25) | |
| Other races | 141 (5.29) | 124 (5.19) | 112 (4.84) | |
| Marital status | | | | 0.66 |
| No | 696 (33.41) | 661 (32.19) | 708 (34.00) | |
| Yes | 1136 (66.59) | 1171 (67.81) | 1125 (66.00) | |
| Education | | | | **0.04** |
| < high school | 807 (39.59) | 757 (39.89) | 760 (39.58) | |
| high school | 695 (35.71) | 714 (33.51) | 764 (39.14) | |
| > high school | 330 (24.70) | 361 (26.60) | 309 (21.29) | |
| BMI, kg/m$^2$ | 33.75 (0.20) | 34.03 (0.18) | 35.15 (0.23) | **< 0.0001** |
| PIR | 2.87 (0.06) | 2.97 (0.07) | 2.86 (0.07) | 0.19 |
| WC, cm | 111.27 (0.39) | 112.33 (0.38) | 115.24 (0.45) | **< 0.0001** |
| ALT, U/L | 32.00 (0.67) | 31.28 (0.68) | 28.73 (0.55) | **0.001** |
| AST, U/L | 28.03 (0.51) | 26.91 (0.49) | 26.64 (0.62) | 0.17 |
| Total cholesterol, mmol/L | 5.19 (0.03) | 5.15 (0.03) | 5.05 (0.04) | **0.01** |
| HDL-cholesterol, mmol/L | 1.23 (0.01) | 1.20 (0.01) | 1.20 (0.01) | **0.03** |
| SII | 311.38 (3.72) | 478.92 (4.24) | 793.72 (10.05) | **< 0.0001** |
| PIV | 131.70 (1.22) | 244.46 (1.11) | 531.10 (8.40) | **< 0.0001** |
| SIRI | 1610.78 (27.36) | 1950.62 (29.49) | 2388.19 (35.09) | **< 0.0001** |
| Drinking | | | | 0.07 |
| Never | 239 (11.06) | 222 (10.34) | 217 (11.29) | |
| Former | 318 (17.51) | 308 (14.28) | 335 (19.00) | |
| Mild | 534 (36.00) | 517 (34.62) | 523 (33.60) | |
| Moderate | 208 (14.64) | 248 (17.52) | 215 (13.54) | |
| Heavy | 353 (20.79) | 401 (23.23) | 382 (22.57) | |
| Smoking | | | | **< 0.0001** |
| No | 1056 (56.80) | 964 (53.54) | 841 (46.25) | |
| Former | 482 (28.33) | 523 (28.38) | 544 (30.89) | |
| Now | 294 (14.87) | 345 (18.08) | 448 (22.86) | |
| Physical work | | | | 0.78 |
| No | 1074 (54.56) | 1064 (52.88) | 1039 (51.80) | |
| Mild | 394 (23.46) | 383 (24.53) | 418 (25.70) | |
| Medium to high | 364 (21.98) | 385 (22.60) | 376 (22.50) | |
| Hypertension | | | | **0.003** |
| No | 896 (51.31) | 856 (49.29) | 749 (44.06) | |
| Yes | 936 (48.69) | 976 (50.71) | 1084 (55.94) | |
| Diabetes | | | | **< 0.001** |
| No | 1266 (74.57) | 1261 (75.68) | 1183 (68.85) | |
| Yes | 566 (25.43) | 571 (24.32) | 650 (31.15) | |

Abbreviations: BMI, body mass index; WC, waist circumference; PIR, family income-to-poverty ratio; HDL, high-density lipoprotein; ALT, alanine transaminase; AST, aspartate transaminase.

## Continuous and categorical associations of SI markers with long-term mortality in FLD

A total of 464 all-cause deaths and 75 cardiovascular deaths were recorded during the follow-up period. Univariate Cox regression indicated that age, BMI, PIR, WC, ALT, total cholesterol, SII, PIV, male sex (with female sex as reference), non-Hispanic Black and non-Hispanic White in ethnicity (with Mexican American as reference), married, education > high school (with < high school as reference), presence of hypertension, presence of diabetes, previous and current alcohol consumption, previous and current smokers, and mild and medium-high physical work were associated with all-cause mortality in patients with FLD; for cardiovascular mortality, age, PIR, total cholesterol, HDL-cholesterol, SII, PIV, presence of hypertension, presence of diabetes, and medium-to-high physical work were significantly associated. Consistently, SIRI was not associated in either all-cause or cardiovascular mortality, and was therefore excluded from further discussion in the multivariate analysis (**S2 Table**).

Multivariate Cox proportional hazards regression analysis demonstrated that SII and PIV (continuous variable) both were significantly associated with all-cause mortality in the adjusted models (HR 1.0004, $P < 0.0001$ for SII and 1.0006, $P = 0.0002$ for PIV). When SII and PIV were treated as categorical variables, both SII and PIV in tertile 3 of the adjusted models (with tertile 1 as the reference) demonstrated significant relevance for all-cause mortality (adjusted model: HR 1.3268, $P = 0.0230$ for SII and HR 1.3212, $P = 0.0357$ for PIV). (**Tables 3 and 4**) Similarly, both SII and PIV were significantly correlated with CVD mortality when treated as continuous variables; however, when treated as categorical variables, PIV (but not SII) was significantly associated with CVD mortality in the adjusted model (HR 2.1350, $P = 0.0307$ for PIV and 1.7462, $P = 0.0824$ for SII) (**S3 Table**).

## Association of SII and PIV with all-cause and CVD-related survival probability in FLD

The KM analysis showed that FLD patients with high tertiles of SII and PIV (compared to low tertiles) both presented lower all-cause related survival (both adjusted $P$ values < 0.0001). However, PIV, but not SII based on tertile was significantly associated with the survival probability from CVD in patients with FLD, with higher PIV having a lower survival probability (adjusted $P = 0.023$), while a similar trend was observed for SII (adjusted $P = 0.069$) (**Fig 2**).

## Potential nonlinear correlation investigation

RCS analysis demonstrated a nonlinear association between SII and all-cause mortality in patients with FLD (turning point = 458.71, $P$ non-linear = 0.0032), while linear correlation was observed

**Table 3. Multivariate adjusted Cox models for SII and all-cause mortality.**

|  | Mortality incidence | Crude Model HR (95%CI) $P$ | Adjusted Model HR (95%CI) $P$ |
|---|---|---|---|
| SII (continuous) | 464/5497 | 1.0006 (1.0004, 1.0008), < **0.0001** | 1.0004 (1.0002, 1.0006), < **0.0001** |
| SII tertile |  |  |  |
| T1 | 129/1832 | 1.0 | 1.0 |
| T2 | 127/1832 | 0.9561 (0.7483, 1.2216), 0.7193 | 0.9012 (0.6888, 1.1789), 0.4477 |
| T3 | 208/1833 | 1.5648 (1.2560, 1.9495), < **0.0001** | 1.3268 (1.0398, 1.6930), **0.0230** |

Abbreviations: BMI, body mass index; PIR, family income-to-poverty ratio. The adjusted model was adjusted for all important covariates in the univariate analysis including age, BMI, PIR, sex, ethnicity, marital status, education level, WC, ALT, total cholesterol, smoking, alcohol consumption, physical work, hypertension, and diabetes.

**Table 4. Multivariate adjusted Cox models for PIV and all-cause mortality.**

|  | Mortality incidence | Crude Model HR (95%CI), *P*-value | Adjusted Model HR (95%CI), *P*-value |
|---|---|---|---|
| PIV (continuous) | 464/5497 | 1.0006 (1.0005, 1.0008), < **0.0001** | 1.0006 (1.0003, 1.0008), **0.0002** |
| PIV tertile |  |  |  |
| T1 | 111/1832 | 1.0 | 1.0 |
| T2 | 146/1832 | 1.3407 (1.0473, 1.7161), **0.0200** | 1.2377 (0.9421, 1.6260), 0.1256 |
| T3 | 207/1833 | 1.9742 (1.5674, 2.4867), < **0.0001** | 1.3212 (1.0188, 1.7134), **0.0357** |

Abbreviations: BMI, body mass index; PIR, family income-to-poverty ratio. The adjusted model was adjusted for all important covariates in the univariate analysis including age, BMI, PIR, sex, ethnicity, marital status, education level, WC, ALT, total cholesterol, smoking, alcohol consumption, physical work, hypertension, and diabetes.

between PIV and all-cause mortality (*P* = 0.2744). (**Fig 3**) Both SII and PIV had a linear correlation with cardiovascular mortality among patients with FLD (both *P* non-linearity > 0.05) (**S1 Fig**).

## Association of SII and PIV with mortality in FLD across subgroups

The *P* for interaction values in all adjusted stratified analyses were > 0.05, suggesting that for SII and PIV, the association with all-cause mortality did not vary across subgroups. (**Fig 4**) Similarly, for cardiovascular mortality, the *P* for interaction values were > 0.05 across all subgroups (**S2 Fig**).

## Interactions with other inflammatory markers, threshold effects, and stability of findings

CRP was similarly positively associated with all-cause and CVD mortality in the FLD population (all-cause: HR 1.089, p = 0.0314; CVD: HR 1.122, *P* = 0.0360). Notably, a significant positive correlation was also observed between SII/PIV and CRP (both β 0.001), suggesting that SII/PIV and other inflammatory markers, such as CRP, may significantly interact and collectively be associated with mortality in the FLD population (**S4** and **S5 Tables**). Threshold effect analysis showed a positive association (HR 1.001) between SII and all-cause mortality in the FLD population after the turning point (SII = 458.71) and a loss of association before the turning point, suggesting the presence of a threshold effect (**S6 Table**). Similar results were obtained when the population with a follow-up period of less than 24 months and extreme SI marker values was not excluded, i.e., SII/PIV remained significantly and positively associated with all-cause and CVD mortality in patients with FLD, confirming the stability of the findings (**S7** and **S8 Tables**).

## Discussion

Using a nationally representative population-based study of complex design and linked to mortality data, we demonstrated that SII and PIV (but not SIRI) were associated with all-cause and CVD mortality. After adjusting for all potential confounding factors, SII and PIV remained associated with all-cause mortality, and their associations did not vary across subgroups. Our results indicated that an increase of 1 unit in SII and PIV was associated with a 0.04% and 0.06% increase in all-cause mortality in FLD patients, respectively. Interestingly, we uncovered a nonlinear association between SII and all-cause mortality, with the turning point of 458.71. Similarly, SII and PIV were significantly associated with cardiovascular mortality in the univariate analysis and multivariate analysis. However, in the survival analysis, we found that PIV, but not SII, was significantly associated with CVD-related survival probabilities over time.

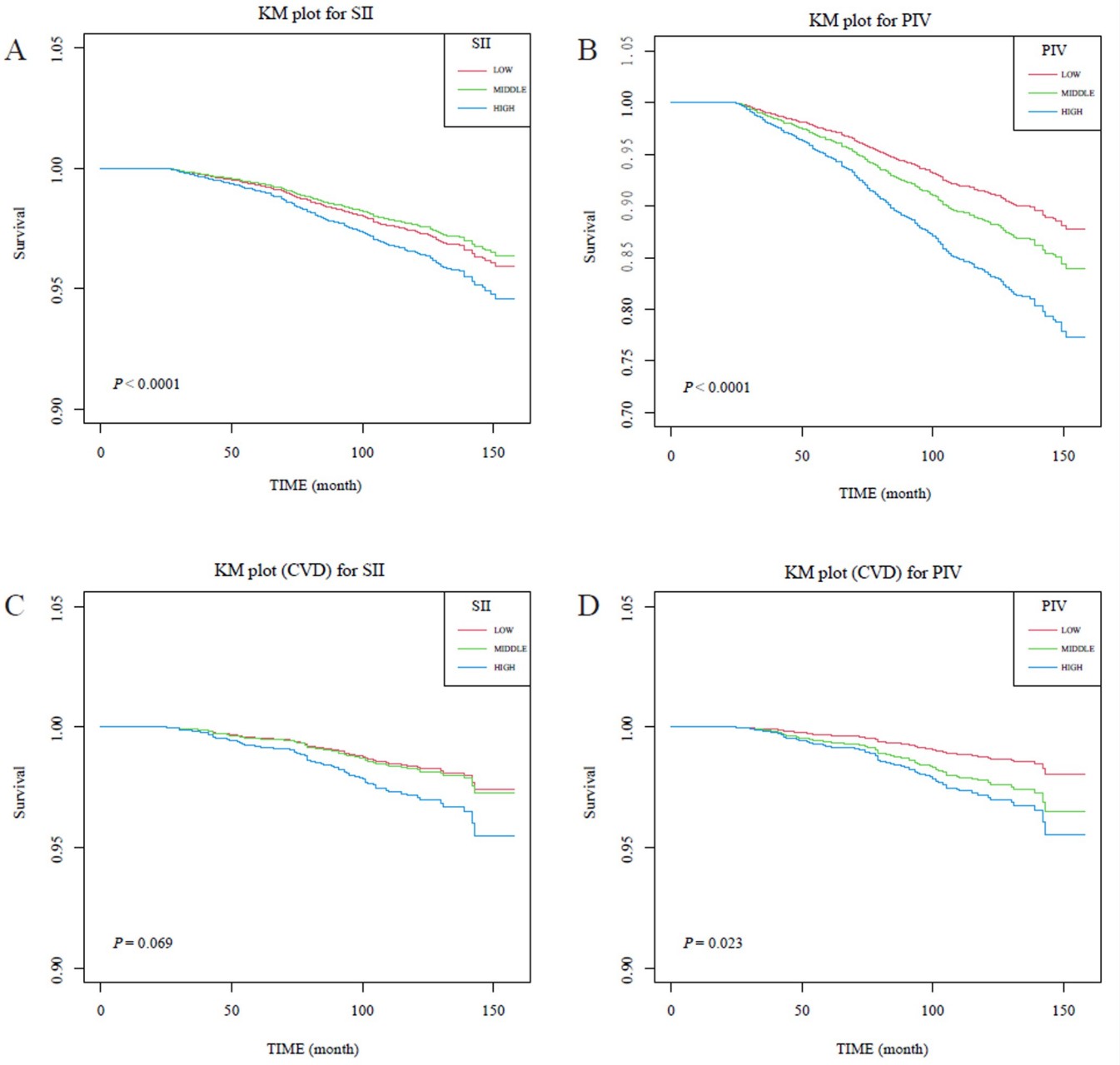

**Fig 2. KM models for SII and PIV with all-cause and cardiovascular mortality in patients with FLD, NHANES 2007–2018.** A: KM plot (all-cause) for SII; B: KM plot (all-cause) for SIRI; C: KM plot (CVD-cause) for SII; D: KM plot (CVD-cause) for PIV. The horizontal axis of the KM plot represents survival time (months) and the vertical axis represents the corresponding (all-cause or CVD) survival probability. The log-rank test was used to analyze differences between groups. The low (T1), middle (T2), and high (T3) tertiles of the SII/PIV were indicated using red, green, and blue solid lines, respectively. Abbreviations: KM, Kaplan-Meier; FLD, fatty liver disease; CVD, cardiovascular disease.

The disease course in patients with FLD can range from simple steatosis to the more severe form of steatohepatitis, which is often closely associated with inflammatory infiltration and fibrosis. Hepatic steatosis, liver injury, and inflammation in NAFLD can promote the development and progression of nonalcoholic steatohepatitis, leading to cirrhosis and decompensated liver events, hepatocellular carcinoma (HCC), and even death.

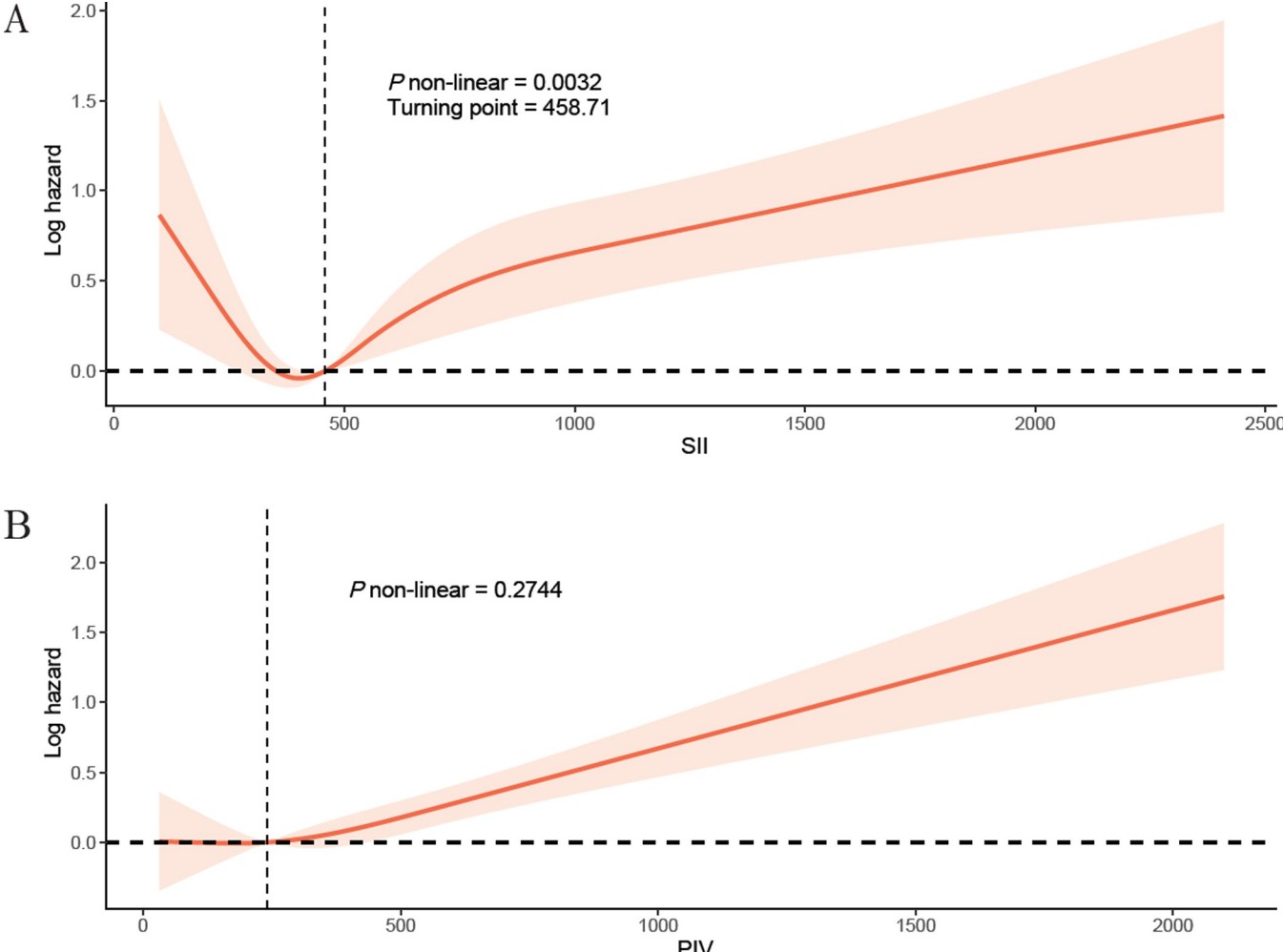

**Fig 3.** RCS models of SII (A) and PIV (B) with all-cause mortality in patients with FLD, NHANES 2007–2018. The horizontal axis of the RCS plot represents the level of SII/PIV, and the vertical axis represents the log hazard value for all-cause mortality. In Panel A, the point where the two dashed lines cross is the turning point (458.71). In panel A, the p non-linear of 0.0032 represented the presence of a non-linear association between SII and all-cause mortality in patients with FLD, while in panel B, the p non-linear was 0.2744, suggesting a linear correlation. Abbreviations: RCS, restricted cubic splines; FLD, fatty liver disease.

The inflammatory context of nonalcoholic steatohepatitis (NASH) is primarily driven by dysregulation of the immune microenvironment and regulated by immune cells, which produce inflammatory mediators that induce lipid overload, oxidative stress, and hepatocyte death. These pathophysiological processes further exacerbate inflammation [30]. These immune cells may promote inflammation directly by secreting inflammatory cytokines or indirectly by activating crosstalk between adjacent immune and non-immune cells [31]. In addition to NAFLD, inflammatory processes also play a role in the progression of AFLD [32]. Specifically, intestinal barrier dysfunction in patients with NAFLD and AFLD leads to bacterial translocation and intestinal dysbiosis, allowing bacteria or bacterial products to reach the liver and circulation to stimulate abnormal immune responses, which are important drivers of inflammation in FLD. Additionally, crosstalk between the liver and adipose tissue can lead to the release of inflammatory mediators that promote liver inflammation and injury [32]. Moreover, since NAFLD/AFLD is closely associated with CVD and systemic low-grade

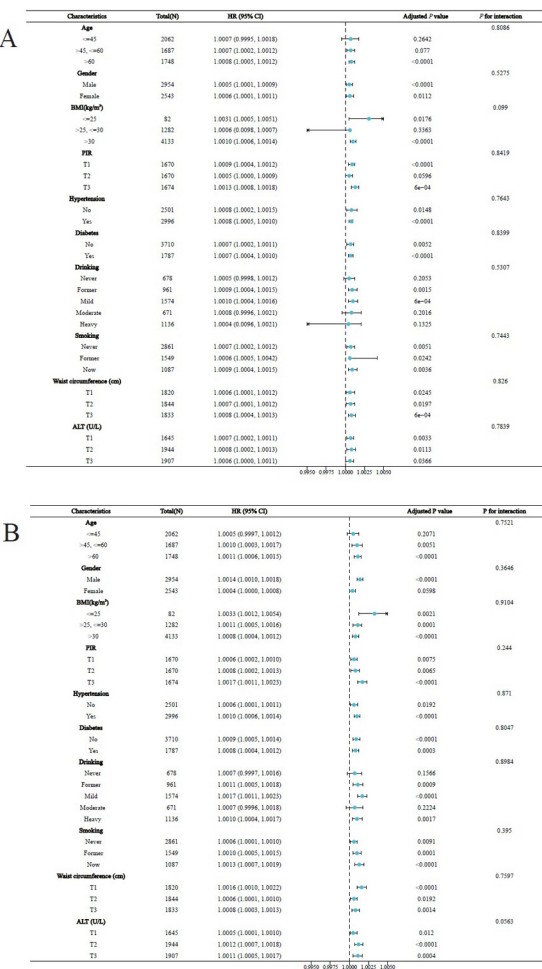

**Fig 4.** Subgroup analysis of SII (A) and PIV (B) with all-cause mortality in patients with FLD, NHANES 2007–2018. In subgroup analyses, "Characteristics" represented the different subgroups included, "Total (N)" represented the number of participants in each subgroup, "HR" represented the effect value of the association in each subgroup (hazard ratio), "Adjusted *P* value" represented the significance of the associations in the subgroups, and "*P* for interaction" represented whether the associations were significantly different between the subgroups. Abbreviations: BMI, body mass index; PIR, family income-to-poverty ratio; ALT, alanine transaminase; T1, T2, T3 = tertile1, tertile2, tertile3.

inflammation is an important shared pathophysiological mechanism, SI markers may also correlate with cardiovascular mortality in patients with FLD [33]. Thus, imbalanced immune response and hepatic inflammation in patients with FLD are major hallmarks of disease progression and characterize the entire course of the disease. Inflammatory markers in the blood of patients with FLD may therefore serve as valuable indicators of disease progression and reflect systemic inflammatory infiltration.

A recent study indicated that SII was associated with hepatic steatosis and exhibited a similar U-shaped positive correlation, but not with hepatic fibrosis, suggesting that systemic inflammation may play a facilitating role in the early progression of NAFLD [16]. In addition, SII is associated with the prognosis of various liver-related outcomes. SII ≥ 330 was associated with a higher recurrence rate and shorter survival after radical resection in patients with HCC and had a higher area under the receiver operating characteristic (ROC) curve compared to other conventional prognostic indicators [20]. A recent study suggested that SII ≥ 610 was a

major survival risk factor in patients with HCC, independent of age [34]. Similarly, SII ≥ 340 was notably predictive of overall survival in patients with HCC [35]. A meta-analysis concluded that SII was significantly associated with overall survival (HR 1.54, $P < 0.001$) and time to recurrence (HR 1.77, $P < 0.001$) in HCC patients [36]. In addition to primary liver tumors, SII was also an independent predictor of recurrence-free survival in those with uveal melanoma liver metastases [37] and overall survival in patients with uveal melanoma liver metastases [38]. Another study revealed that preoperative SII ≥ 870 was independently associated with 90-day mortality in postoperative liver transplant patients [39]. Using the NHANES database, SII exhibited a U-shaped positive correlation with all-cause and cause-specific mortality in patients with CVD [40] and hypertension [41], similar to our findings. Recent cohort studies similarly utilizing NHANES have shown that SII is associated with mortality in chronic kidney disease [42], osteoarthritis [43], and diabetes [44] populations. Most previous studies have similarly demonstrated nonlinear associations (e.g., U-shaped associations) between SII and clinical outcomes. These results suggest that there is a threshold effect for these associations, i.e., only excessive SI promotes disease progression in FLD. Mild inflammatory responses may exert hepatoprotective effects by promoting tissue repair and homeostatic reconstruction [45]. However, the underlying biological mechanisms explaining why a certain range of SII is negatively associated with all-cause mortality in patients with FLD remain unclear.

A recent study involving 133 consecutive children and adolescents aged 6–18 years with obesity found that PIV was independently associated with the presence and severity of hepatic steatosis [46]. Furthermore, PIV has been widely implicated in the prognosis of multiple cancers [47], patients on peritoneal dialysis [48], hypertensive patients [15], sepsis patients [49], patients with ST elevation myocardial infarction [50], and those with with acute heart failure [51]. Our findings suggested that higher tertile PIV was positively associated with all-cause and CVD mortality and that PIV (but not SII) was linked to CVD-related survival probabilities in survival analyses. Since PIV takes into account the number of monocytes in the blood compared to SII, it is plausible that monocytes may have a greater impact on CVD-related prognosis. A study indicated that serum CC-chemokine ligand 2 (CCL2) was associated with comorbidities but not liver disease progression in patients with NAFLD, and that CCL2 could direct blood monocytes to damaged or injured tissues, suggesting that considering monocytes in addition to SII may increase the prognostic value for CVD [52]. Given the current lack of literature describing the prognostic value of PIV in patients with FLD, the underlying mechanisms warrant further exploration.

Interestingly, we did not establish the prognostic value of SIRI in patients with FLD, although SIRI was related to all-cause and cardiovascular mortality in the general and hypertensive populations in NHANES [13,14]. SIRI has been shown to be associated with a poor immune checkpoint inhibitor response and a compromised immune microenvironment in patients with HCC [53]. Furthermore, SIRI has been identified as an independent predictor of survival in patients with HCC [54], cholangiocarcinoma [55], and gallbladder cancer [56]. A recent study revealed that SIRI was related with metabolic disorders and CVD (ORs of 2.092, 3.441, 1.417, 2.080, and 3.397 for metabolic disorders, metabolic syndrome, dyslipidemia, central obesity, and 10-year CVD risk in the high tertile of SIRI compared to the low tertile of SIRI, respectively) [57]. Evidence to explain the absence of an association between SIRI and mortality in patients with FLD is currently lacking. Since SIRI differs from the other two SI markers primarily by not incorporating platelet counts, we speculate that platelet counts may have important prognostic value in patients with FLD. Interestingly, a recent meta-analysis demonstrated that reduced platelet counts were associated with increased NAFLD severity [58]. In the future, more studies are needed, to elucidate the predictive and prognostic value of SIRI in patients with FLD.

There are several important advantages to our research. Firstly, we used a national population-based database and obtained mortality data to reduce bias and enhance generalizability. Secondly, we explored three novel SI markers and elucidated their association with mortality in patients with FLD through systematic prognostic analyses including baseline data, univariate and multivariate analysis, survival analysis, RCS modelling, and stratified analysis. In addition, we have adjusted for possible confounding variables in the analysis to ensure the validity of the effect values if possible. However, our study still has limitations. Firstly, due to the nature of this study was an observational study, we were not able to address causality well. Therefore, these findings may only indicate an association between these SI markers and mortality in patients with FLD, rather than suggesting that SI directly contributes to increased mortality and needs to be interpreted with caution. Secondly, residual confounding factors may still not have been included. In addition, patients with FLD are diagnosed based on FLI instead of liver biopsy and therefore may lack accuracy. Due to the data limitations of NHANES, future studies should assess FLD using more accurate methods such as transient elastography to accurately reflect these associations. However, due to various limitations of liver biopsy such as poor scalability and potential procedural concerns, and the established accuracy of FLI, it can be a relatively good indicator of the presence of FLD. Liver biopsy, being invasive and subject to potential sampling errors and lack of inter-rater reliability, is impractical for large epidemiologic surveys [59]. Additionally, a recent meta-analysis demonstrated that FLI had the highest predictive ability for hepatic steatosis compared to other non-invasive markers [60]. Future studies are needed to further explore the potential of SI markers as prognostic indicators in the FLD population.

## Conclusions

In a population-based retrospective cohort study, we found significant associations between SII and PIV and all-cause mortality in patients with FLD, with a nonlinear association for SII. PIV may be more closely associated with the survival probability over time linked to CVD. Further clinical research will be needed to confirm our findings in diverse populations and ethnicities, and to utilize other diagnostic tools for fatty liver disease, such as transient elastography. Additionally, further investigation into the potential of these markers as prognostic indicators in patients with fatty liver disease is warranted. These findings may aid in the stratification of mortality risk among individuals with fatty liver disease.

## Supporting information

**S1 Table. Baseline characteristics and linked mortality data of patients according to SIRI tertile.**
(DOCX)

**S2 Table. Univariate analysis for all-cause mortality and cardiovascular mortality.**
(DOCX)

**S3 Table. Multivariate Cox models for SII and PIV and cardiovascular mortality.**
(DOCX)

**S4 Table. Association of CRP with all-cause and CVD mortality in patients with FLD, NHANES 2007–2018.**
(DOCX)

**S5 Table. Correlation analysis of SII/PIV with CRP.**
(DOCX)

**S6 Table. Threshold effects analysis of the association of SII with all-cause mortality in the FLD population.**
(DOCX)

**S7 Table. Sensitivity analyses of the association of SII/PIV with all-cause mortality in patients with FLD (excluding those with <24 months of follow-up and extreme values of SI markers).**
(DOCX)

**S8 Table. Sensitivity analyses of the association of SII/PIV with CVD mortality in patients with FLD (excluding those with <24 months of follow-up and extreme values of SI markers).**
(DOCX)

**S1 Fig.** RCS models of SII (A) and PIV (B) with cardiovascular mortality in patients with FLD, NHANES 2007–2018. The horizontal axis of the RCS plot represents the level of SII/PIV, and the vertical axis represents the log hazard value for CVD mortality.
(DOCX)

**S2 Fig.** Subgroup analysis of SII (A) and PIV (B) with cardiovascular mortality in patients with FLD, NHANES 2007–2018.
(DOCX)

**S1 Data.**
(CSV)

## Author Contributions

**Conceptualization:** Chenglin Sun, Hongpeng Guo.

**Data curation:** Xinghe Pan, Rui Zhang, Hongpeng Guo.

**Formal analysis:** Xinghe Pan, Jie Lv, Hongpeng Guo.

**Investigation:** You Li, Yitong Zhang.

**Methodology:** You Li, Yitong Zhang, Junliang Liu.

**Resources:** Rui Zhang.

**Software:** You Li, Yitong Zhang, Rui Zhang, Junliang Liu.

**Supervision:** Chenglin Sun, Hongpeng Guo.

**Validation:** Rui Zhang, Junliang Liu.

**Visualization:** Junliang Liu.

**Writing – original draft:** Xinghe Pan, Jie Lv, Man Liu, You Li, Hongpeng Guo.

**Writing – review & editing:** Chenglin Sun, Hongpeng Guo.

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
