## [Decision Letter · Decision Letter 0]

4 Sep 2024

PONE-D-24-25755Chronic systemic inflammation predicts long-term mortality among patients with fatty liver disease: Data from the National Health and Nutrition Examination Survey 2007-2018PLOS ONE

Dear Dr. hongpeng,

Thank you for submitting your manuscript to PLOS ONE. After careful consideration, we feel that it has merit but does not fully meet PLOS ONE’s publication criteria as it currently stands. Therefore, we invite you to submit a revised version of the manuscript that addresses the points raised during the review process.

We look forward to receiving your revised manuscript.

Kind regards,

Andreas Zirlik, MD

Academic Editor

PLOS ONE

Journal Requirements:

2. Thank you for stating the following financial disclosure: This study were supported under the Liaoning Provincial Science and Technology Plan project (Grant no.2022JH2/1013),the Shenyang Science and Technology Plan Project(Grant no.21173918),the Graduate Student Science and Technology Innovation Fund of Shenyang Medical College (Grant no. Y20220531) and Shenyang Health Commission Scientific Research Project (Grant no. 202358).

Reviewers' comments:

Reviewer's Responses to Questions

**Comments to the Author**

1. Is the manuscript technically sound, and do the data support the conclusions?

Reviewer #1: Yes

Reviewer #2: Partly

2. Has the statistical analysis been performed appropriately and rigorously? 

Reviewer #1: No

Reviewer #2: No

3. Have the authors made all data underlying the findings in their manuscript fully available?

Reviewer #1: Yes

Reviewer #2: Yes

4. Is the manuscript presented in an intelligible fashion and written in standard English?

Reviewer #1: No

Reviewer #2: Yes

5. Review Comments to the Author

Reviewer #1: In this manuscript Pan et al. present interesting data from the NHANES cohort evaluating Pan et al. on the associations of chronic systemic inflammation with long-term mortality among patients with fatty liver disease. While this manuscript shows interesting data, there are several points related to the analytical approach and the presentation of the data that remain to be substantially improved.

Major comments:

1. Section “Abstract”: Please provide more details regarding the design (in brief) of the NHANES cohort and on the collection of follow-up information. Moreover, the description of the statistical methods used and the presentation of results used appears confusing and therefore should be carefully revised. For example, since Cox proportional hazard models are frequently used for time-to event analyses, including survival analysis, the difference between Cox models and survival analyses remains unclear based on the information in the abstract.

2. Section “Methods”: The description of the NHANES cohort should be improved, to allow a better understanding of the population/eligibility criteria in NHANES. Please also cite corresponding references regarding the study design of the NHANSE cohort for some guidance.

3. Section “Methods”: The methods are very hard to understand. The section should be substantially revised to provide a clear understand of the methods used.

4. Section “Methods”: While the use of stepwise models may be of advantage for co-variate adjustment in datasets with small samples, stepwise regression models appear not ideal to analyze the associations between markers of inflammation and clinical outcomes in the present dataset. Cox proportional hazard models or semiparametric models should be preferred, the sample size allows for a comprehensive list of covariates without the need for any stepwise approach.

5. Section “Methods”: Please explain the choice of the covariates used for adjustment in model 2. It is unclear why a set of recognized predictors of adverse clinical events were not used. Please also list the co-variates used in model 3.

6. Section “Result”: The term “linked mortality data” in Table 1 is confusion. The risk for clinical outcomes according to indices of systemic inflammation should be presented in Tables 3+4 while providing incidence rates per 100-patient years in addition to the crude and adjusted HRs.

7. Section “Results”: Please structure the section based on the implications of systemic inflammation makers (e.g. Baseline characteristics, associations of different markers with clinical outcomes), not on the statistical method used. The findings for all associations should be described by all corresponding models (categorical, continuous, Poission, interaction), respectively.

8. Section “Results”: The indices of systemic inflammation are based on counts of platelet/neutrophil/monocyte count/lymphocyte count. How do the findings relate to inflammatory biomarkers such as IL-6 or hsCRP? Please add the analyses of the associations of additional inflammatory biomarkers with clinical outcomes.

Minor comments:

1. Section “Methods”: The paragraph summarizing the covariable selection can be streamlined.

2. Section “Results”: Please list incidence rates or estimates when referring to the risk for clinical outcomes. P-values alone do not provide any useful information.

3. Supplemental Tables: Uni- and multivariate analyses can be shown for multiple outcomes in one table.

Reviewer #2: The present manuscript titled "Chronic systemic inflammation predicts long-term mortality among patients with fatty liver disease: Data from the National Health and Nutrition Examination Survey 2007-2018" explores the relationship between systemic inflammation markers (SII, PIV, and SIRI) and mortality outcomes in patients with fatty liver disease (FLD). The study utilizes NHANES data, applying statistical methods, including multivariate Cox regression models, Kaplan-Meier survival curves, and restricted cubic splines. While the manuscript addresses a significant clinical question and the methods are generally well-chosen, several methodological and clarity issues need attention. These include potential biases introduced by the exclusion of certain patients and extreme values, and inconsistencies in covariate adjustment across models. Additionally, the terminology used to describe statistical relationships and the clarity of figure legends require improvement. Addressing these concerns will enhance the manuscript’s validity, clarity, and overall contribution to the field.

Major Concerns

• The term "U-shaped positive correlation" used in the manuscript is somewhat misleading. It implies that the correlation remains consistently positive, which is not typically the case with a U-shaped relationship, where the direction of the relationship changes. This terminology could cause confusion for readers. The authors should clarify the mathematical calculation underlying this term or explain how they arrived at this specific terminology. Alternatively, the authors could reference the "turning point" in the discussion to provide a more accurate description of the relationship. A clearer explanation will help avoid misunderstandings and ensure that the findings are accurately interpreted.

• The exclusion of patients who died within 24 months may introduce bias, as it could exclude individuals with the most severe disease. This exclusion criterion should be better justified or re-evaluated. In addition, the authors explain, that “we excluded extreme values of SI markers (N=70)“. Can the authors elaborate what they mean by that? How was “extreme” defined? Which cut offs were used and why? Excluding values of SI markers could also introduce bias, as it might remove data points that could be crucial for understanding the full range of the relationship between SI markers and mortality.

• The manuscript employs different approaches to adjusting for covariates in all-cause and cardiovascular mortality models. While age is a critical risk factor for mortality, it appears that age was not consistently adjusted for in the all-cause mortality models. This inconsistency can introduce significant bias and affect the validity of the study's conclusions. The authors should either provide a strong justification for this approach or revise their models to ensure that age is consistently adjusted for across all mortality analyses. This is crucial to maintain the methodological integrity of the study and the accuracy of its findings.

• Building up on the previous comment, the authors demonstrate significant differences in possibly confounding factors such as age, BMI, Sex, and others across groups. These factors should be consistently adjusted for in the statistical models to ensure accurate results. If these covariates were not included in all models, the authors should provide a better explanation of how they selected the covariates for adjustment. The process of covariate selection is not clear to this reviewer, and without proper justification, there is a risk of introducing bias or misinterpreting the relationships between variables.

• The retrospective design inherently limits the ability to infer causality. While this is common in cohort studies, the discussion could better address how this limitation might impact the findings.

• The study relies heavily on NHANES data, which, while comprehensive, limits the analysis to the diagnostic tools and variables available within that dataset. The discussion could benefit from considering how additional diagnostic tools, might enhance future research.

• The present study uses the Fatty Liver Index (FLI) as a diagnostic tool for FLD rather than liver biopsy, which may reduce diagnostic accuracy. While the authors acknowledge this limitation, they could have provided more justification or compared the FLI's diagnostic reliability with other methods in the discussion.

• The manuscript introduces SIRI as one of the systemic inflammation markers but then downplays its relevance due to non-significant findings. This could be better integrated into the discussion by exploring why SIRI might not have shown significant associations, contrasting with SII and PIV.

• The U-shaped association between SII and mortality is intriguing but not fully explored in the discussion. The manuscript could expand on potential biological mechanisms underlying this relationship, offering more than just statistical interpretation.

• Some figures, particularly the Kaplan-Meier curves and restricted cubic splines, are essential to understanding the findings. However, the figure legends provided in the manuscript are too brief, lacking detailed explanations. Each figure, especially those with multiple panels marked by letters (e.g., A, B, C, etc.), should include a comprehensive description. This should detail the meaning of each axis if abbreviations are used, and the statistical tests employed. Providing this information will enhance the clarity of the figures and ensure that readers can fully understand the visual data without needing to reference the main text.

Minor Concerns

• The abstract and introduction sections are well-written but could be slightly condensed to improve focus. The statement "Future clinical research is needed to confirm these findings in diverse populations and with different diagnostic tools like transient elastography" is quite general and does not add specific value to the abstract. It is recommended to remove this statement to maintain the focus and precision of the abstract.

• While the manuscript is generally well-cited, a few statements, particularly those regarding the general prognostic value of SII and PIV in other diseases, could benefit from additional references or more current studies.

• Both the terms "gender" and "sex" are used inconsistently throughout the manuscript, which can lead to confusion. To ensure clarity and scientific accuracy, it is important to standardize the terminology. If referring to biological differences, "sex" should be used consistently throughout the manuscript. Ensuring uniformity in terminology will improve the overall coherence and precision of the text.

6. PLOS authors have the option to publish the peer review history of their article (what does this mean?). If published, this will include your full peer review and any attached files.

Reviewer #1: No

Reviewer #2: No

---

## [Author Response · Author response to Decision Letter 0]

10 Sep 2024

Dear Editors and Reviewers,

Thank you very much for your great efforts and patience in reviewing the manuscript and for giving us the opportunity to make revisions. We have responded to all the reviewers' comments and revised the manuscript accordingly. We believe that the quality of our manuscript will be significantly improved after the revision. The following is our response.

Response to reviewer 1

Major comments:

1. Section “Abstract”: Please provide more details regarding the design (in brief) of the NHANES cohort and on the collection of follow-up information. Moreover, the description of the statistical methods used and the presentation of results used appears confusing and therefore should be carefully revised. For example, since Cox proportional hazard models are frequently used for time-to event analyses, including survival analysis, the difference between Cox models and survival analyses remains unclear based on the information in the abstract.

RESPONSE: We appreciate your constructive and careful suggestions. We strongly agree with you and have added more details of NHANES' cohort design to the abstract as follows: “Baseline FLD participants from NHANES 2007-2018 were follow-up until December 31, 2019, and were prospectively matched with records in the National Death Index database to obtain mortality information.” In addition, we have revised the description of the statistical analyses in the abstract to address the ambiguous portions in response to your comments, and the revised counterparts are as follows: “Univariate and multivariate Cox proportional hazards models were used to estimate hazard ratios for mortality outcomes. Kaplan-Meier (KM) analysis was used to estimate survival probabilities, and restricted cubic splines (RCS) were employed to explore potential nonlinear associations. Stratified analysis was conducted to assess the robustness of the findings across different subgroups.”

2.Section “Methods”: The description of the NHANES cohort should be improved, to allow a better understanding of the population/eligibility criteria in NHANES. Please also cite corresponding references regarding the study design of the NHANSE cohort for some guidance.

RESPONSE: We appreciate your constructive and careful suggestions. We have supplemented the description of the NHANES cohort in the Methods section with citations to the corresponding literature based on your comments, as follows: “NHANES is a nationally representative, multistage, cross-sectional survey, which is a major program of the National Center for Health Statistics (NCHS) and aimed to assess the health and nutrition status of US non-institutionalized populations. The NCHS Ethics Review Board has approved all NHANES study protocols, and all participants signed a written consent form. We adopted data from NHANES 2007-2018 (including 6 continuous cycles, 2007-2008, 2009-2010, 2011-2012, 2013-2014, 2015-2016, 2017-2018) for this study. Baseline FLD participants from NHANES 2007-2018 were followed until December 31, 2019, and mortality information was obtained by prospectively matching with death certificate information from the Public-use Linked Mortality Files of the National Death Index (NDI) database (https://www.cdc.gov/nchs/data-linkage/mortality-public.htm)[13, 18].”

References

13. Xia Y, Xia C, Wu L, Li Z, Li H, Zhang J. Systemic Immune Inflammation Index (SII), System Inflammation Response Index (SIRI) and Risk of All-Cause Mortality and Cardiovascular Mortality: A 20-Year Follow-Up Cohort Study of 42,875 US Adults. J Clin Med. 2023;12(3). doi: 10.3390/jcm12031128.

18. Liu Y, Geng T, Wan Z, Lu Q, Zhang X, Qiu Z, et al. Associations of Serum Folate and Vitamin B12 Levels With Cardiovascular Disease Mortality Among Patients With Type 2 Diabetes. JAMA Netw Open. 2022;5(1):e2146124. doi: 10.1001/jamanetworkopen.2021.46124.

3. Section “Methods”: The methods are very hard to understand. The section should be substantially revised to provide a clear understand of the methods used.

RESPONSE: We appreciate your constructive and detailed suggestions regarding the clarity of the Methods section. Based on your comments, we have made substantial revisions to improve the logical flow and structure of this section. Below are the specific modifications:Participants Section:We have added more context on the design of NHANES, including the fact that it is a nationally representative, multistage, cross-sectional survey led by the National Center for Health Statistics (NCHS). This clarifies the nature of the population sampled and the methodology behind NHANES.We provided a detailed explanation of the exclusion criteria used to derive the final sample of 5,497 participants. This includes the removal of individuals without FLD diagnostic indicators, those with missing mortality data, missing covariate information, and extreme values for systemic inflammation (SI) markers. These steps clarify how we arrived at the final population.We clarified the follow-up process, specifying that FLD participants were followed until December 31, 2019, and mortality information was obtained by matching with the National Death Index (NDI) database.Definition of SI Markers:We retained the original definitions of the SI markers (SII, SIRI, PIV) but added more detail regarding how they were calculated using data from complete blood counts (CBC). This makes it clear that the markers are derived from specific blood parameters.Definition of FLD:We retained the use of the Fatty Liver Index (FLI) to define hepatic steatosis and included a detailed breakdown of the formula used to calculate FLI. This ensures a clear understanding of how FLD was diagnosed in the study.Outcome Ascertainment:We clarified how mortality data was obtained, specifying that it was matched with death certificates in the NDI database and coded according to ICD-10 classifications for cardiovascular deaths. This ensures transparency in the outcome data collection process. Covariate Determination:We have provided a more detailed description of the key confounding factors used in the models, such as smoking, alcohol consumption, and physical activity, and how these factors were categorized. This step helps to better explain the variables adjusted for in our analysis.Statistical Analysis:We have restructured the statistical analysis section to clearly differentiate between univariate and multivariate Cox proportional hazards models. We now explicitly state which covariates were included in the fully adjusted models for both all-cause and cardiovascular mortality.We added a more detailed explanation of Kaplan-Meier survival curves and log-rank tests, and how these were used to estimate survival probabilities across different levels of SI markers. This further clarifies the survival analysis techniques used.We explained the use of restricted cubic splines (RCS) for exploring nonlinear relationships and described how the degrees of freedom (knots) were determined based on statistical significance. This addresses the more complex statistical analyses performed in the study.Stratified analysis and sensitivity analysis details were expanded, including how we tested the stability of our results across different subgroups and how other inflammatory markers, such as C-reactive protein (CRP), were analyzed in relation to SII and PIV. This provides a more complete understanding of the robustness checks we performed.We added a detailed description of the sensitivity analyses, particularly focusing on the inclusion of participants with shorter follow-up times (<24 months) and extreme SI values to assess the robustness of the results.We explained the threshold effect analysis of SII on all-cause mortality, which explores the nonlinear association and potential cutoff points for increased mortality risk. This further strengthens our findings.

4.Section “Methods”: While the use of stepwise models may be of advantage for co-variate adjustment in datasets with small samples, stepwise regression models appear not ideal to analyze the associations between markers of inflammation and clinical outcomes in the present dataset. Cox proportional hazard models or semiparametric models should be preferred, the sample size allows for a comprehensive list of covariates without the need for any stepwise approach.

RESPONSE: We strongly agree with your comment that Cox proportional hazards models or semiparametric models are more suitable for analyzing the associations between markers of inflammation and clinical outcomes in our dataset, given the sample size and the comprehensive list of covariates.we have made the following changes:Removal of Stepwise Regression Models: We have removed the description of stepwise regression from the Methods section, as well as the stepwise multiple regression table in the Supplementary Material. We agree that the sample size allows us to include a full range of covariates in the Cox proportional hazards models without resorting to stepwise selection.Replacement with Cox Proportional Hazards Models: We now rely solely on Cox proportional hazards models for multivariable adjustment, ensuring a more robust and interpretable analysis. All covariates that were considered important based on prior literature and clinical relevance have been included in the models without any stepwise reduction.We appreciate your constructive and insightful suggestions regarding the use of stepwise regression models. 

5.Section “Methods”: Please explain the choice of the covariates used for adjustment in model 2. It is unclear why a set of recognized predictors of adverse clinical events were not used. Please also list the co-variates used in model 3.

RESPONSE: We appreciate your constructive and careful suggestions. We couldn't agree more with your suggestion. For the selection of covariates for the previous partial adjustment of Model 2, we picked certain underlying covariates based on previous studies similarly using NHANES ((e.g. PMID: 36817427, doi:10.3389/fimmu.2023.1087345), i.e., demographic variables such as age. Thus, Model 2 was originally constructed to further adjust the underlying demographic variables based on the crude model in order to observe changes in effect sizes and to derive whether the association was independent of simple demographic variables. However, in light of your comments, we have deleted the original Model 2 and retained only the crude and fully adjusted models to more clearly reflect the process of adjusting for covariates and to reduce confusion about covariate selection. Finally, we have added covariates to Model 3 (fully adjusted model) at your request as follows: “For all-cause mortality, crude model was an unadjusted crude model, and the fully adjusted model was adjusted for all important covariates in the univariate analysis including age, BMI, PIR, sex, ethnicity, marital status, education level, WC, ALT, total cholesterol, smoking, alcohol consumption, physical work, hypertension, and diabetes. For cardiovascular mortality, crude model was a crude model, and the fully adjusted model was adjusted for all significant covariates including age, PIR, total cholesterol, HDL- cholesterol, physical work, hypertension, and diabetes.”

6.Section “Result”: The term “linked mortality data” in Table 1 is confusion. The risk for clinical outcomes according to indices of systemic inflammation should be presented in Tables 3+4 while providing incidence rates per 100-patient years in addition to the crude and adjusted HRs.

RESPONSE: We appreciate your constructive and careful suggestions. We have deleted the inappropriate descriptions in Table 1 and added descriptions of incidence rates to Tables 3/4 as requested.

Table 3. Multivariate adjusted Cox models for SII and all-cause mortality.

 Crude Model HR (95%CI) P Fully adjusted Model HR (95%CI) P

SII (continuous) (mortality incidence: 464/5497)

 1.0006 (1.0004, 1.0008), < 0.0001 1.0004 (1.0002, 1.0006), < 0.0001

SII tertile 

T1 (129/1832) 1.0 1.0

T2 (127/1832)

 0.9561 (0.7483, 1.2216), 0.7193 0.9012 (0.6888, 1.1789), 0.4477

T3 (208/1833) 1.5648 (1.2560, 1.9495), < 0.0001 1.3268 (1.0398, 1.6930), 0.0230

Abbreviations: BMI, body mass index; PIR, family income-to-poverty ratio. The fully adjusted model was adjusted for all important covariates in the univariate analysis including age, BMI, PIR, sex, ethnicity, marital status, education level, WC, ALT, total cholesterol, smoking, alcohol consumption, physical work, hypertension, and diabetes.

Table 4. Multivariate adjusted Cox models for PIV and all-cause mortality.

 Crude Model HR (95%CI), P-value Fully adjusted Model HR (95%CI), P-value

PIV (continuous) (mortality incidence: 464/5497) 1.0006 (1.0005, 1.0008), < 0.0001 1.0006 (1.0003, 1.0008), 0.0002

PIV tertile 

T1 (111/1832) 1.0 1.0

T2 (146/1832) 1.3407 (1.0473, 1.7161), 0.0200 1.2377 (0.9421, 1.6260), 0.1256

T3 (207/1833) 1.9742 (1.5674, 2.4867), < 0.0001 1.3212 (1.0188, 1.7134), 0.0357

Abbreviations: BMI, body mass index; PIR, family income-to-poverty ratio. The fully adjusted model was adjusted for all important covariates in the univariate analysis including age, BMI, PIR, sex, ethnicity, marital status, education level, WC, ALT, total cholesterol, smoking, alcohol consumption, physical work, hypertension, and diabetes.

7.Section “Results”: Please structure the section based on the implications of systemic inflammation makers (e.g. Baseline characteristics, associations of different markers with clinical outcomes), not on the statistical method used. The findings for all associations should be described by all corresponding models (categorical, continuous, Poission, interaction), respectively.

RESPONSE: We appreciate your constructive and careful suggestions. We apologize for the distress caused by the inappropriate description. We have substantially revised the Methods section to make it clearer, as indicated in the Results section of the revised manuscript.

8.Section “Results”: The indices of systemic inflammation are based on counts of platelet/neutrophil/monocyte count/lymphocyte count. How do the findings relate to inflammatory biomarkers such as IL-6 or hsCRP? Please add the analyses of the associations of additional inflammatory biomarkers with clinical outcomes.

RESPONSE: We appreciate your constructive and careful suggestions. We strongly agree with your suggestion and have added the appropriate analyses. Since NHANES only has data on CRP and not on other inflammatory markers such as IL-6, we explored the association of CRP with mortality in patients with FLD and elucidated the correlation between SII/PIV and CRP.We added the corresponding sensitivity analysis section in the manuscript: “In sensitivity analyses, we first explored the association of other inflammatory markers such as C-reactive protein (CRP) with mortality in patients with FLD and the correlation of CRP with SII/PIV.”, “CRP was similarly positively associated with all-cause and CVD mortality in the FLD population (all-cause: HR 1.089, p=0.0314; CVD: HR 1.122, p=0.0360). Notably, there was also a significant positive correlation between SII/PIV and CRP (both β 0.001), suggesting that SII/PIV and other inflammatory markers, such as CRP, may significantly interact and collectively be associated with mortality in the FLD population (Table S4 and S5).”

Minor comments:

1. Section “Methods”: The paragraph summarizing the covariable selection can be streamlined.

RESPONSE: We appreciate your constructive and careful suggestions. We simplified the covariate selection section based on comments as follows: “We selected several important confounding factors including age, sex, ethnicity, education level, family income-to-poverty ratio (PIR), and marital status, smoking, alcohol consumption, physical work, BMI, WC, total cholesterol, high-density lipoprotein (HDL)-cholesterol, alanine transaminase (ALT), aspartate transaminase (AST), hypertension, and diabetes. We defined types of smoking as: (1) Never: individuals who answered that they had smoked less than 100 cigarettes in life; (2) Former: have smoked at least 100 cigarettes throughout their lives, but are currently not smoking; (3) Current: had smoked at least 100 cigarettes in life and smoked at present[24]. We determined types of drinking

---

## [Decision Letter · Decision Letter 1]

1 Oct 2024

PONE-D-24-25755R1Chronic systemic inflammation predicts long-term mortality among patients with fatty liver disease: Data from the National Health and Nutrition Examination Survey 2007-2018PLOS ONE

Dear Dr. hongpeng,

Thank you for submitting your manuscript to PLOS ONE. After careful consideration, we feel that it has merit and significantly imporved though revision but does not fully meet PLOS ONE’s publication criteria as it currently stands. Therefore, we invite you to submit a revised version of the manuscript that addresses the points raised during the review process (see Reviewer 1). 

We look forward to receiving your revised manuscript.

Kind regards,

Andreas Zirlik, MD

Academic Editor

PLOS ONE

Journal Requirements:

Reviewers' comments:

Reviewer's Responses to Questions

**Comments to the Author**

1. If the authors have adequately addressed your comments raised in a previous round of review and you feel that this manuscript is now acceptable for publication, you may indicate that here to bypass the “Comments to the Author” section, enter your conflict of interest statement in the “Confidential to Editor” section, and submit your "Accept" recommendation.

Reviewer #1: (No Response)

Reviewer #2: All comments have been addressed

2. Is the manuscript technically sound, and do the data support the conclusions?

Reviewer #1: Yes

Reviewer #2: Yes

3. Has the statistical analysis been performed appropriately and rigorously? 

Reviewer #1: Yes

Reviewer #2: Yes

4. Have the authors made all data underlying the findings in their manuscript fully available?

Reviewer #1: Yes

Reviewer #2: Yes

5. Is the manuscript presented in an intelligible fashion and written in standard English?

Reviewer #1: No

Reviewer #2: Yes

6. Review Comments to the Author

Reviewer #1: In the revised version of the manuscript, Pan et al. have overall addressed the points raised in my previous comments. However, some points, which are mainly related to the presentation of the data, formatting, and grammar remain to be addressed.

Major comments:

1. Section “Abstract”: While the author's effort is appreciated, the newly added content is way too long for the abstract. My initial comment only asks for more clear wording related to the survival analyses, not for a comprehensive summary of statistical methods (this should be described in the “Methods” section). Since there is no formal need to name all methods used in detail, it would be sufficient (instead of the newly included language) to state that the associations between chronic systemic inflammation and long-term mortality were analyzed (following the description of the study design). Please streamline the “Methods” section in the abstract accordingly.

2. Section “Methods”, paragraph “Participants”: This section should be renamed to something like “Study design”. Accordingly, the authors should only provide a description of the overall NHANES cohort, and inclusion and exclusion criteria for the present study. The study flow should be moved to “Results”, where it should be listed in a paragraph labeled “Participants”. Please also add the median duration of follow-up.

3. Section “Results”, “Baseline characteristics”: Since you are comparing data across groups of increasing inflammatory burden - please replace the current p-values (based on the methods paragraph, the present numbers seem to be overall-p-values) with p-values for trend. Please also indicate in the heading of the column that p-values for trends are used.

4. Section “Results”, “Tables 3+4”: In the current version, the newly added event numbers are not clearly labeled and added in the first column (which should only list the variables). Please add a separate column for the newly incorporated event numbers, and please also make sure to include an appropriate heading.

5. Section“Results”: Despite my previous comment, the results section and its headings are still structured based on the statistical methods used. In this current version, it is hard to follow the logic. To improve the structure, please organize the sections in a way that is consistent with common standards for the reporting of clinical research. Regarding this manuscript, use sections such as participants, baseline characteristics, categorical and continuous associations between markers of chronic systemic inflammation and long-term mortality… In other words, the logic of reporting should be organized based on paragraphs related to findings, not according to the methods. Please structure the section and its headings accordingly.

6.

Minor comments:

1. While the revised manuscript appears approved, it still contains multiple sections that are in parts hard to read due to grammar issues. Please revise the grammar and syntax of the whole manuscript to ensure appropriate readability. English language editing is highly recommended.

2. If only crude and one adjusted model are used, there is no need to call the adjusted model fully adjusted, since the is no other model adjusted for fewer covariates.

3. The figure legends are in part not comprehensive enough. Please review and improve to ensure a clear wording.

Reviewer #2: (No Response)

7. PLOS authors have the option to publish the peer review history of their article (what does this mean?). If published, this will include your full peer review and any attached files.

Reviewer #1: No

Reviewer #2: No

---

## [Author Response · Author response to Decision Letter 1]

2 Oct 2024

Dear Editors and Reviewers,

We sincerely appreciate your time, effort, and valuable feedback in reviewing our manuscript. Your thoughtful comments have given us the opportunity to improve our work, and we have carefully addressed each of the reviewers' suggestions. We believe that the revisions have significantly enhanced the quality of the manuscript. Please find below our detailed responses to the reviewers' comments.

Thank you once again for your guidance and consideration.

Response to reviewer 1

Major comments:

1. Section “Abstract”: While the author's effort is appreciated, the newly added content is way too long for the abstract. My initial comment only asks for more clear wording related to the survival analyses, not for a comprehensive summary of statistical methods (this should be described in the “Methods” section). Since there is no formal need to name all methods used in detail, it would be sufficient (instead of the newly included language) to state that the associations between chronic systemic inflammation and long-term mortality were analyzed (following the description of the study design). Please streamline the “Methods” section in the abstract accordingly.

RESPONSE: We appreciate your constructive and careful suggestions.We strongly agree with your comments and have streamlined the “Methods” section of the abstract in line with the suggestions, and the revised Methods section was as follows: “Methods: This was a retrospective cohort study. We included adult patients with FLD with complete data and analyzed the association between chronic SI and long-term mortality in patients with FLD. Systemic immune-inflammation index (SII), pan-immune-inflammation value (PIV), and systemic inflammation response index (SIRI) were evaluated based on peripheral blood counts and FLD was determined by the Fatty Liver Index.”.

2. Section “Methods”, paragraph “Participants”: This section should be renamed to something like “Study design”. Accordingly, the authors should only provide a description of the overall NHANES cohort, and inclusion and exclusion criteria for the present study. The study flow should be moved to “Results”, where it should be listed in a paragraph labeled “Participants”. Please also add the median duration of follow-up.

RESPONSE: We appreciate your constructive and careful suggestions.We strongly agree with your comments and have made one-to-one modifications in accordance with your modification instructions. We added a description of the follow-up duration to the “Participants” section of the results as requested. The corresponding sections of the modified methods and results were as follows:

METHODS

Study design

NHANES is a nationally representative, multistage, cross-sectional survey conducted by the National Center for Health Statistics (NCHS) and is designed to assess the health and nutrition status of US non-institutionalized populations. The NCHS Ethics Review Board approved all NHANES study protocols, and all participants provided written informed consent. We utilized data from NHANES 2007-2018 (including 6 continuous cycles, 2007-2008, 2009-2010, 2011-2012, 2013-2014, 2015-2016, 2017-2018) for this study. Baseline FLD participants from NHANES 2007-2018 were followed until December 31, 2019, and mortality data was obtained by prospectively matching with death certificate information from the Public-use Linked Mortality Files of the National Death Index (NDI) database (https://www.cdc.gov/nchs/data-linkage/mortality-public.htm)[13, 18]. The study inclusion criteria included FLD patients aged 18 years or older with complete data. Exclusion criteria were participants with missing data and extreme SI values and a follow-up duration of less than 24 months.

RESULTS

Participants

We initially enrolled 57,414 individuals aged 18 years and older and excluded those without diagnostic indicators for FLD (N=40,125). Next, we excluded individuals with missing mortality data (N=2325), missing blood biochemical tests data (N=54), missing covariate data such as marital status (N=721), education level (N=14), hypertension (N=137), smoking (N=12), and physical work (N=2). Finally, from the population with complete data, we excluded extreme values of SI markers (N=70), those without FLD according to the diagnostic criteria (N=7832), and patients who died within 24 months of follow-up (N=626). Therefore, 5497 patients with FLD (2954 men and 2543 women) were included in the current study. (Figure 1) The median follow-up duration for FLD patients was 88 months(interquartile range: 55-121 months).

3.Section “Results”, “Baseline characteristics”: Since you are comparing data across groups of increasing inflammatory burden - please replace the current p-values (based on the methods paragraph, the present numbers seem to be overall-p-values) with p-values for trend. Please also indicate in the heading of the column that p-values for trends are used.

RESPONSE: We appreciate your constructive and careful suggestions.We strongly agree with your comment and have changed the original p-value to “p-values for trends” in the corresponding section of the results and table 1/2. Please see the corresponding revised section of the manuscript.

4.Section “Results”, “Tables 3+4”: In the current version, the newly added event numbers are not clearly labeled and added in the first column (which should only list the variables). Please add a separate column for the newly incorporated event numbers, and please also make sure to include an appropriate heading.

RESPONSE: We appreciate your constructive and careful suggestions.We strongly agree with your comments and have added an additional column to Table 3/4 to describe the mortality incidence overall and in the different tertiles. Please see revised table 3/4 below:

Table 3. Multivariate adjusted Cox models for SII and all-cause mortality.

 Mortality incidence Crude Model HR (95%CI) P Adjusted Model HR (95%CI) P

SII (continuous)

 464/5497 1.0006 (1.0004, 1.0008), < 0.0001 1.0004 (1.0002, 1.0006), < 0.0001

SII tertile 

T1 129/1832 1.0 1.0

T2

 127/1832 0.9561 (0.7483, 1.2216), 0.7193 0.9012 (0.6888, 1.1789), 0.4477

T3 208/1833 1.5648 (1.2560, 1.9495), < 0.0001 1.3268 (1.0398, 1.6930), 0.0230

Abbreviations: BMI, body mass index; PIR, family income-to-poverty ratio. The adjusted model was adjusted for all important covariates in the univariate analysis including age, BMI, PIR, sex, ethnicity, marital status, education level, WC, ALT, total cholesterol, smoking, alcohol consumption, physical work, hypertension, and diabetes.

Table 4. Multivariate adjusted Cox models for PIV and all-cause mortality.

 Mortality incidence Crude Model HR (95%CI), P-value Adjusted Model HR (95%CI), P-value

PIV (continuous) 464/5497 1.0006 (1.0005, 1.0008), < 0.0001 1.0006 (1.0003, 1.0008), 0.0002

PIV tertile 

T1 111/1832 1.0 1.0

T2 146/1832 1.3407 (1.0473, 1.7161), 0.0200 1.2377 (0.9421, 1.6260), 0.1256

T3 207/1833 1.9742 (1.5674, 2.4867), < 0.0001 1.3212 (1.0188, 1.7134), 0.0357

Abbreviations: BMI, body mass index; PIR, family income-to-poverty ratio. The adjusted model was adjusted for all important covariates in the univariate analysis including age, BMI, PIR, sex, ethnicity, marital status, education level, WC, ALT, total cholesterol, smoking, alcohol consumption, physical work, hypertension, and diabetes.

5. Section“Results”: Despite my previous comment, the results section and its headings are still structured based on the statistical methods used. In this current version, it is hard to follow the logic. To improve the structure, please organize the sections in a way that is consistent with common standards for the reporting of clinical research. Regarding this manuscript, use sections such as participants, baseline characteristics, categorical and continuous associations between markers of chronic systemic inflammation and long-term mortality… In other words, the logic of reporting should be organized based on paragraphs related to findings, not according to the methods. Please structure the section and its headings accordingly.

RESPONSE: We appreciate your constructive and careful suggestions.We strongly agree with your suggestions and apologize for the unsuitability of the previous revisions. In response to your comments, we have changed the headings of the Results section accordingly to reflect the specific content of the findings. Please see section headings below:

Participants

We initially enrolled 57,414 individuals aged 18 years and older and excluded those without diagnostic indicators for FLD (N=40,125). Next, we excluded individuals with missing mortality data (N=2325), missing blood biochemical tests data (N=54), missing covariate data such as marital status (N=721), education level (N=14), hypertension (N=137), smoking (N=12), and physical work (N=2). Finally, from the population with complete data, we excluded extreme values of SI markers (N=70), those without FLD according to the diagnostic criteria (N=7832), and patients who died within 24 months of follow-up (N=626). Therefore, 5497 patients with FLD (2954 men and 2543 women) were included in the current study. (Figure 1) The median follow-up duration for FLD patients was 88 months(interquartile range: 55-121 months).

Baseline characteristics 

Table 1 described the baseline characteristics of patients grouped according to SII tertiles (≤ 370.9, 370.9 ~ 571.8, and ≥ 571.8). BMI, WC, ALT, AST, SII, PIV, SIRI, sex, and ethnicity showed significant differences between groups (all p-values for trend < 0.0001). Statistically significant trends were also observed for total cholesterol, hypertension, diabetes, and smoking (all p-values for trend < 0.05). However, no significant differences were observed between groups in age, PIR, HDL-cholesterol, marital status, education level, alcohol consumption, and physical work (all p-values for trend > 0.05). Table 2 presented the baseline characteristics of patients grouped according to PIV tertiles (≤ 184.4, 184.4 ~ 313.8, and ≥ 313.8). Significant differences were observed in BMI, SII, PIV, SIRI, ethnicity, and smoking (all p-values for trend < 0.0001), and in age, ALT, total cholesterol, HDL-cholesterol, education, hypertension, and diabetes between groups (all p-values for trend < 0.05). However, no significant differences were found for PIR, AST, sex, marital status, alcohol consumption, and physical work (all p-values for trend > 0.05). Baseline analysis grouped by tertile of SIRI was presented in Table S1.

Continuous and categorical associations of SI markers with long-term mortality in FLD

A total of 464 all-cause deaths and 75 cardiovascular deaths were recorded during the follow-up period. Univariate Cox regression indicated that age, BMI, PIR, WC, ALT, total cholesterol, SII, PIV, male sex (with female sex as reference), non-Hispanic Black and non-Hispanic White in ethnicity (with Mexican American as reference), married, education > high school (with < high school as reference), presence of hypertension, presence of diabetes, previous and current alcohol consumption, previous and current smokers, and mild and medium-high physical work were associated with all-cause mortality in patients with FLD; for cardiovascular mortality, age, PIR, total cholesterol, HDL-cholesterol, SII, PIV, presence of hypertension, presence of diabetes, and medium-to-high physical work were significantly associated. Consistently, SIRI was not associated in either all-cause or cardiovascular mortality, and was therefore excluded from further discussion in the multivariate analysis. (Table S2)

Multivariate Cox proportional hazards regression analysis demonstrated that SII and PIV (continuous variable) both were significantly associated with all-cause mortality in the adjusted models (HR 1.0004, P < 0.0001 for SII and 1.0006, P = 0.0002 for PIV). When SII and PIV were treated as categorical variables, both SII and PIV in tertile 3 of the adjusted models (with tertile 1 as the reference) demonstrated significant relevance for all-cause mortality (adjusted model: HR 1.3268, P = 0.0230 for SII and HR 1.3212, P = 0.0357 for PIV). (Table 3 and 4) Similarly, both SII and PIV were significantly correlated with CVD mortality when treated as continuous variables; however, when treated as categorical variables, PIV (but not SII) was significantly associated with CVD mortality in the adjusted model (HR 2.1350, P = 0.0307 for PIV and 1.7462, P = 0.0824 for SII). (Table S3)

Association of SII and PIV with all-cause and CVD-related survival probability in FLD

The KM analysis showed that FLD patients with high tertiles of SII and PIV (compared to low tertiles) both presented lower all-cause related survival (both adjusted P values < 0.0001). However, PIV, but not SII based on tertile was significantly associated with the survival probability from CVD in patients with FLD, with higher PIV having a lower survival probability (adjusted P = 0.023), while a similar trend was observed for SII (adjusted P = 0.069). (Figure 2)

Potential nonlinear correlation investigation

RCS analysis demonstrated a nonlinear association between SII and all-cause mortality in patients with FLD (turning point = 458.71, P non-linear = 0.0032), while linear correlation was observed between PIV and all-cause mortality (P = 0.2744). (Figure 3) Both SII and PIV had a linear correlation with cardiovascular mortality among patients with FLD (both P non-linearity > 0.05). (Figure S1)

Association of SII and PIV with mortality in FLD across subgroups

The P for interaction values in all adjusted stratified analyses were > 0.05, suggesting that for SII and PIV, the association with all-cause mortality did not vary across subgroups. (Figure 4) Similarly, for cardiovascular mortality, the P for interaction values were > 0.05 across all subgroups. (Figure S2)

Interactions with other inflammatory markers, threshold effects, and stability of findings

CRP was similarly positively associated with all-cause and CVD mortality in the FLD population (all-cause: HR 1.089, p=0.0314; CVD: HR 1.122, P=0.0360). Notably, a significant positive correlation was also observed between SII/PIV and CRP (both β 0.001), suggesting that SII/PIV and other inflammatory markers, such as CRP, may significantly interact and collectively be associated with mortality in the FLD population (Table S4 and S5). Threshold effect analysis showed a positive association (HR 1.001) between SII and all-cause mortality in the FLD population after the turning point (SII=458.71) and a loss of association before the turning point, suggesting the presence of a threshold effect (Table S6). Similar results were obtained when the population with a follow-up period of less than 24 months and extreme SI marker values was not excluded, i.e., SII/PIV remained significantly and positively associated with all-cause and CVD mortality in patients with FLD, confirming the stability of the findings (Table S7 and S8).

Minor comments:

1. While the revised manuscript appears approved, it still contains multiple sections that are in parts hard to read due to grammar issues. Please revise the grammar and syntax of the whole manuscript to ensure appropriate readability. English language editing is highly recommended.

RESPONSE: We appreciate your constructive and careful suggestions.We strongly agree with your comments and have revised the full text for grammar and fluency. We invited specialized professionals to professionally edit the revised manuscript in the English language.

2. If only crude and one adjusted model are used, there is no need to call the adjusted model fully adjusted, since the is no other model adjusted for fewer covariates.

RESPONSE: We appreciate your constructive and careful suggestions.We strongly agree with your comments and have deleted “fully” before adjusted model throughout the text to accurately reflect the model adjustments.

3. The figure legends are in part not comprehensive enough. Please review and improve to ensure a clear wording.

RESPONSE: We appreciate your constructive and careful suggestions.We strongly agree with your comments and have added and improved the figure legend. The revised figure legend was as follows:

Figure legend

Figure 1. Study subject inclusion and exclusion flow

---

## [Editor Report · Decision Letter 2]

15 Oct 2024

Chronic systemic inflammation predicts long-term mortality among patients with fatty liver disease: Data from the National Health and Nutrition Examination Survey 2007-2018

PONE-D-24-25755R2

Dear Dr. hongpeng,

We’re pleased to inform you that your manuscript has been judged scientifically suitable for publication and will be formally accepted for publication once it meets all outstanding technical requirements.

Kind regards,

Andreas Zirlik, MD

Academic Editor

PLOS ONE
---

## [Editor Report · Acceptance letter]

7 Nov 2024

PONE-D-24-25755R2 

PLOS ONE

Dear Dr. hongpeng, 

I'm pleased to inform you that your manuscript has been deemed suitable for publication in PLOS ONE. Congratulations! Your manuscript is now being handed over to our production team.

Kind regards, 

on behalf of

Univ. Prof. Dr. Andreas Zirlik 

Academic Editor

PLOS ONE